# Beyond Tip of the Iceberg: A Simple-yet-Effective Approach to Class Imbalanced Semi-supervised Node Classification

## ABSTRACT

Graph Neural Networks (GNNs) have achieved great success in dealing with non-Euclidean graph-structured data and have been widely deployed in many real-world applications. However, their effectiveness is often jeopardized under class-imbalanced training sets. Most existing studies have analyzed class-imbalanced node classification from a supervised learning perspective, they do not fully utilize the large number of unlabeled nodes in semi-supervised scenarios. We claim that the supervised signal is just the tip of the iceberg and a large number of unlabeled nodes have not yet been effectively utilized. In this work, we propose `IceBerg`, a debiased self-training framework to address the class-imbalanced and few-shot challenges for GNNs at the same time. Specifically, to figure out the Matthew effect and label distribution shift in self-training, we propose `Double Balancing`, which can largely improve the performance of existing baselines with just a few lines of code as a simple plug-and-play module. Secondly, to enhance the long-range propagation capability of GNNs, we disentangle the propagation and transformation operations of GNNs. Therefore, the weak supervision signals can propagate more effectively to address the few-shot issue. In summary, we find that leveraging unlabeled nodes can significantly enhance the performance of GNNs in class-imbalanced and few-shot scenarios, and even small, surgical modifications can lead to substantial performance improvements. Systematic experiments on benchmark datasets show that our method can deliver considerable performance gain over existing class-imbalanced node classification baselines. Additionally, due to `IceBerg`'s outstanding ability to leverage unsupervised signals, it also achieves state-of-the-art results in few-shot node classification scenarios. The code of `IceBerg` is available at: https://anonymous.4open.science/r/IceBerg-D865/.

## CCS CONCEPTS

• **Computing methodologies → Artificial intelligence**.

## KEYWORDS

Graph Neural Networks; Semi-Supervised Learning; Long-tail

**ACM Reference Format:**
Anonymous Author(s). 2024. Beyond Tip of the Iceberg: A Simple-yet-Effective Approach to Class Imbalanced Semi-supervised Node Classification. In *Proceedings of ACM Conference (Conference'17)*. ACM, New York, NY, USA, 9 pages. https://doi.org/10.1145/nnnnnnn.nnnnnnn

## 1 INTRODUCTION

Semi-supervised node classification is a fundamental task in graph machine learning, holding significant relevance in various real-world applications, such as fraud detection [6, 20], recommendation [34], and credit risk prediction [30] to name some. With the rapid development of deep learning, Graph Neural Networks (GNNs) have been widely used in dealing with non-Euclidean graph-structured data and have achieved considerable progress. However, their effectiveness is often jeopardized under class-imbalanced training datasets. In these scenarios, GNNs are prone to be biased toward majority classes, leading to low test accuracy on minority classes.

To tackle the class-imbalanced issues in deep learning, various Class Imbalanced Learning (CIL) methods have been proposed in fields like computer vision and natural language processing [3, 12, 16]. However, these methods are hard to directly apply to graph-structured data because of the non-iid characteristics of graphs [27]. Recently, close to the heels of the rapid development of GNNs in node classification, various Class Imbalanced Graph Learning (CIGL) approaches have been proposed [23, 26, 27, 42], most of them attempt to utilize data augmentation techniques (*e.g.*, SMOTE, Mixup, and GAN) to generate virtual minority nodes for balancing the training process [23, 24, 42]. The other line of approaches aims to facilitate CIGL through the graph structure. More precisely, they adjust margins node-wisely according to the extent of deviation from connectivity patterns or augment structures to alleviate ambivalent and distant message passing. Nevertheless, most of the existing methods treat the CIGL task as supervised learning, overlooking the large amount of unlabeled data in the graph. For example, in the Cora dataset, if there are 20 labels per majority class, and the step imbalance ratio equals 10 (which means there are only 2 labels per minority class), labeled nodes in the majority classes for only 4.4% of all nodes, while in the minority classes, it is even lower at just 0.6%. Therefore, we naturally raise a question:

*"Can we explicitly utilize these unlabeled nodes to assist with CIGL?"*

Self-training is one of the most promising Semi-Supervised Learning (SSL) paradigms for bypassing the labeling cost by leveraging abundant unlabeled data. It typically selects a subset of predictions from the model itself with a higher confidence of correctness as pseudo labels to add to the training set and repeats this process iteratively. However, traditional self-training methods are based on a basic assumption that the class distribution of the training set is balanced, and the class imbalanced issue can be more problematic for self-training algorithms. We first analyze this phenomenon from the quantity and confidence of pseudo labels in a class imbalanced training dataset (as shown in Figure 1 Left). We can observe that: (1) Due to the abundance of training labels for majority classes, the model is prone to be biased toward majority classes, resulting in a much higher amount of majority class pseudo labels compared to minority classes. (2) Because of the imbalance in the model training process, the model tends to be more confident in its predictions for

majority classes, and vice versa. If we directly use self-training algorithms on class imbalanced training sets, the predefined threshold will filter out most pseudo labels from minority classes, resulting in an increasingly imbalanced training set. Therefore, with the increase in stages, the model's performance will deteriorate further (as shown in Figure 1 Right), and we refer to this phenomenon as the *Matthew Effect* ("the rich get richer and the poor get poorer").

Although some previous works have attempted to use self-training to aid with learning in CIGL [37, 40, 43], they only use pseudo labels to fill minority classes and do not fully leverage unlabeled nodes. Additionally, their methods are all based on multi-stage framework, which requires multiple rounds of teacher-student distillation, and this will significantly reduce the efficiency of the model. So we aim to design an approach within the single-stage self-training framework, but we still face several challenges. First, since the class distribution between the labeled and unlabeled set is inconsistent (as shown in Figure 2), even if the supervised training on the labeled set is balanced, the pseudo labels generated by the model will still be imbalanced, which will also lead to the *Matthew Effect*. Second, because the ground-truth labels of unlabeled set are unavailable, the class distribution of unlabeled set is unknown, so we are hard to conduct CIGL on the unlabeled set. Surprisingly, we found that the pseudo labels generated by the model can serve as a good estimation of unlabeled set class distribution, enabling us to perform CIGL on the pseudo label set. Consequently, we propose a simple-yet-effective self-training method, `Double Balancing` (DB), which only requires a few lines of code to the existing pipelines and can significantly improve the model's performance in CIGL with almost no additional training overhead. Specifically, we first use the model to predict pseudo labels for the unlabeled nodes, then use them to estimate the class distribution of the unlabeled set, and finally apply a simple balanced loss function to mitigate the imbalance issue. Due to the potential presence of incorrect predictions in the pseudo labels, which may lead to *Confirmation Bias* [1], we propose `Noise-Tolerant Double Balancing` to further enhance performance.

Additionally, most previous CIGL baselines have only focused on the balance between majority and minority classes, without addressing the potential few-shot problem in CIGL. Therefore, we revisit the model architecture from the perspective of message-passing. Though message-passing is a key factor of GNNs for capturing structural knowledge, such a coupled design may in turn bring severe challenges for GNNs when learning with scarce labels [19]. Because of the scarcity of labeled nodes in minority classes, limited propagation steps make it difficult for the supervision signals to cover unlabeled nodes. However, simply increasing the model depth will result in the over-smoothing problem, which we refer to as *Propagation Dilemma*. According to previous literature [41], the major cause for the performance degradation of deep GNNs is the model degradation issues caused by a large number of transformations, rather than a large number of propagation. We decouple GNNs and increase the propagation hops, interestingly, we find that increasing the number of propagation hops can effectively enhance CIGL performance, even surpassing some existing specific CIGL baselines. In summary, we claim that the advantages of decouple GNNs in CIGL: (1) By increasing the number of propagation hops, we can transmit the supervision signals further, capturing higher-order structural

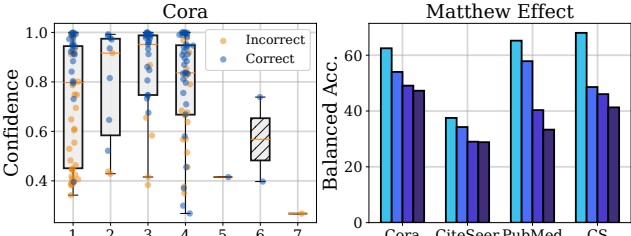

**Figure 1:** *Left*: **Confidence of pseudo labels in the long-tailed node classification task. The first four classes are majority classes, and the latter three are minority classes.** *Right*: **Matthew Effect of standard multi-stage self-training on long-tailed graph datasets.**

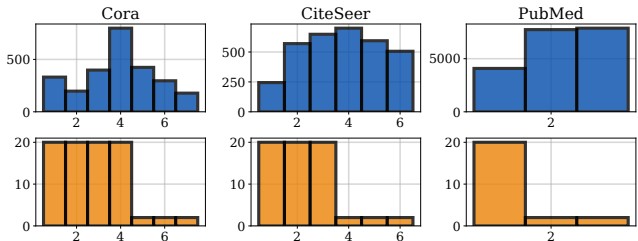

**Figure 2: The first row is the class distribution of the unlabeled set, and the second row is that of the labeled set.**

knowledge, and thereby alleviating the few-shot problem. (2) Since the feature propagation process can be pre-computed and does not participate in model training, the efficiency of the model will be significantly enhanced. (3) Because the feature propagation can be considered as unsupervised representation learning, the noise introduced by incorrect labels will not backpropagate to the node features [4, 36], which will be more beneficial for self-training.

Combining all the above designs, we propose `IceBerg`, a simple-yet-effective approach for class-imbalanced and few-shot node classification. Our contributions can be listed as follows:

- **Preliminary Analysis.** We believe that supervision signals are just the tip of the iceberg. By effectively leveraging the large number of unlabeled nodes in the graph, we can easily and significantly enhance the model's performance.
- **Model Design.** Based on our preliminary analysis, we propose `IceBerg`, a simple-yet-effective approach. It can also be flexibly combined with other baselines as a plug-and-play module.
- **Experimental Evaluation.** Systematic and extensive experiments demonstrate that `IceBerg` achieves superior performance across various datasets and experimental settings in CIGL. Additionally, in light of the strong few-shot ability of `IceBerg`, it can also obtain state-of-the-art performance in few-shot scenarios.
- **Benchmark Development.** We integrate diverse backbones, datasets, baselines, and experimental settings in our repository. Researchers can evaluate all combinations with less effort.

## 2 PRELIMINARY

**Notations.** To maintain consistency of notations, we use bold uppercase and lowercase letters to represent matrices and vectors, and calligraphic font types to denote sets. Given an attributed graph denoted as $\mathcal{G} = (\mathcal{V}, \mathbf{A}, \mathbf{X})$, where $\mathcal{V} = \{v_1, v_2, \ldots, v_N\}$ is the set of $N$ nodes; we denote the adjacency matrix as $\mathbf{A} \in \mathbb{R}^{N \times N}$, if $v_i$ and $v_j$ are connected, $\mathbf{A}_{ij} = 1$, otherwise $\mathbf{A}_{ij} = 0$; $\mathbf{X} = [\mathbf{x}_1, \mathbf{x}_2, \ldots, \mathbf{x}_N] \in \mathbb{R}^{N \times D}$ is the node feature matrix, each node $v_i$ is associated with a $D$-dimensional node feature vector $\mathbf{x}_i$. The normalized adjacency matrix is represented by $\tilde{\mathbf{A}} = \mathbf{D}^{-1/2} \mathbf{A} \mathbf{D}^{-1/2}$, where $\mathbf{D} \in \mathbb{R}^{N \times N}$ is a diagonal degree matrix $\mathbf{D}_{ii} = \sum_j \mathbf{A}_{ij}$.

**Graph Neural Networks.** Following the diagram of Message-Passing Neural Networks (MPNNs), most forward processes of MPNNs can be defined as:

$$\mathbf{m}_i^{(l-1)} = \text{PROPAGATE}\left(\{\mathbf{h}_i^{(l-1)}, \mathbf{h}_j^{(l-1)} | j \in \mathcal{N}(i)\}\right), \quad (1)$$

$$\mathbf{h}_i^{(l)} = \text{TRANSFORM}\left(\mathbf{m}_i^{(l-1)}\right), \quad (2)$$

where $\mathbf{h}_i^{(l)}$ is the feature vector of node $v_i$ in the $l$-th layer and $\mathbf{m}_i^{(l-1)}$ is the aggregated message vector from the $(l-1)$-th layer, $\mathcal{N}(i)$ is a set of neighbor nodes of node $v_i$. PROPAGATE (P) denotes the message-passing function of aggregating neighbor information, and TRANSFORM (T) denotes the non-linear mapping with node features as input. According to the ording the model arrages the P and T operations, we can roughly classify the existing GNN architectures into three categories: PTPT, PPTT, and TTPP [41]. Typical GNNs, such as GCN, GAT, and GraphSAGE, entangle P and T in each layer, so they can be classified as PTPT. PPTT architecture disentangles P and T, and stacks multiple P in preprocessing. While TTPP also disentangles two operations, but embed node features first by T and then the stacked P can be considered as label propagation.

**Long-Tailed Semi-supervised Node Classification.** In this task, we have a labeled node set $\mathcal{V}_L \subset \mathcal{V}$ and unlabeled node set $\mathcal{V}_U = \mathcal{V} \setminus \mathcal{V}_L$. The target of the node classification task is to train a model $f_\theta$ based on labeled nodes $\mathcal{V}_L$ to predict the classes of unlabeled nodes $\mathcal{V}_U$. For the labeled node $v_i$, it is associated with a ground-truth label $\mathbf{y}_i \in \{0, 1\}^C$, where $C$ is the number of classes. Let $N_L^c$ denote the number of samples for class $c$ in the labeled dataset, and $N_U^c$ denote the number of samples for class $c$ in the unlabeled dataset. We have $N_L^1 \geq N_L^2 \geq \ldots N_L^C$, and the imbalance ratio of the labeled dataset is denoted by $\gamma_L = \frac{N_L^1}{N_L^C}$. Similarly, the imbalance ratio of the unlabeled dataset is $\gamma_U = \frac{\max_c N_U^c}{\min_c N_U^c}$.

## 3 METHODOLOGY

In this section, we will give a detailed description of IceBerg. We first introduce our debiased self-training method, Double Balancing, in Section 3.1. And then, we revisit model architecture from message-passing in Section 3.2. Finally, we introduce the overall framework of IceBerg and explain the advantages of combining the two modules mentioned above.

## 3.1 Double Balancing

We start with the formulation of the self-training methods by analyzing the corresponding loss function. Many existing self-training methods seek to minimize a supervised classification loss on labeled data and an unsupervised loss on unlabeled data. Formally, the objective function is given as follows:

$$\min_{\theta \in \Theta} \mathcal{L}_{ssl} = \underbrace{\mathbb{E}_{v_i \in \mathcal{V}_L} \ell(f_\theta(v_i), y_i)}_{\text{supervised}}$$
$$+ \underbrace{\lambda \cdot \mathbb{E}_{v_j \in \mathcal{V}_U} \mathbb{I}(\max(f_\theta(v_j)) \geq \tau) \ell(f_\theta(v_j), \hat{y}_j)}_{\text{unsupervised}}, \quad (3)$$

where $\ell$ is Cross-Entropy (CE) loss, $\lambda$ is a trade-off hyper-parameter to balance supervised and unsupervised loss, $\mathbb{I}(\cdot)$ is an indicator function, $\tau$ is the confidence threshold, and $\hat{y}$ is the prediction generated by model $f_\theta$. If we analyze CIGL from the perspective of domain adaptation [10], we can treat the labeled and unlabeled set as the source domain and the test set as the target domain, therefore we can rewrite Equation 3 as follows to estimate the test error:

$$\text{error} = \mathbb{E}\ell(f_\theta(v_i), y_i)\frac{p_t(\mathbf{x}, y)}{p_l(\mathbf{x}, y)} + \lambda \cdot \mathbb{E}\ell(f_\theta(v_j), \hat{y}_j)\frac{p_t(\mathbf{x}, y)}{p_u(\mathbf{x}, y)}$$
$$= \mathbb{E}\ell(f_\theta(v_i), y_i)\frac{p_t(y)p_t(\mathbf{x}|y)}{p_l(y)p_l(\mathbf{x}|y)} + \lambda \cdot \mathbb{E}\ell(f_\theta(v_j), \hat{y}_j)\frac{p_t(y)p_t(\mathbf{x}|y)}{p_u(y)p_u(\mathbf{x}|y)} \quad (4)$$

where $p_l(\mathbf{x}, y)$, $p_u(\mathbf{x}, y)$, and $p_t(\mathbf{x}, y)$ represents data distribution of labeled set, unlabeled set, and test set respectively. $p(y)$ and $p(x|y)$ are class distribution and class conditional distribution. For simplicity, we omit the subscripts of expectation. We assume the class conditional distribution of labeled, unlabeled, and test sets are consistent here, namely $p_l(\mathbf{x}|y) = p_u(\mathbf{x}|y) = p_t(\mathbf{x}|y)$ (actually this assumption does not always hold, and we will explain it in the next section). Since the target test class distribution is balanced, and the source labeled class distribution is imbalanced, we can consider CIGL as a label distribution shift problem [7, 9], where $p_l(y) \neq p_t(y)$. Existing CIGL methods aim to make $p_l(y)$ close to $p_t(y)$ using the known class distribution, in order to force the model becomes unbiased.

However, if we analyze the unsupervised term, we will find that the aforementioned approach does not work because $p_u(y)$ is unknown. Furthermore, by observing the pseudo labels predicted by the model on class imbalanced datasets, we can find that: (1) because the model is biased toward majority classes, it is prone to generate majority pseudo labels. (2) Since the decision boundary is far from the majority classes, resulting in higher confidence for the majority classes (as shown in Figure 1 Left). This means that if we do not conduct operations on the unsupervised term, traditional self-training algorithms will become increasingly imbalanced, leading to the emergence of the *Matthew Effect* (as shown in Figure 1 Right). So if the model training is balanced, *i.e.* we assume that the model has consistent pseudo labeling ability across all classes, can we solve this problem? Unfortunately, the answer is negative. As shown in Figure 2, in real-world scenarios, the costs of data collection and labeling vary across different classes, and the unlabeled data may not be balance distributed. Additionally, the class distribution between labeled and unlabeled sets could be inconsistent, which

we refer to as label *Missing Not At Random*. Therefore, even if the model has consistent pseudo labeling capability for each class, the generated pseudo labels may still be imbalanced.

According to the above analysis, we attempt to conduct `Double Balancing` for the unsupervised term. Surprisingly, due to the high accuracy of pseudo labels, they can serve as a good estimate of pseudo class distribution. First, we utilize the model to generate pseudo labels for unlabeled nodes, and we count the number of pseudo labels across all classes:

$$\tau' = \frac{1}{|\mathcal{V}_U|} \sum_j^{|\mathcal{V}_U|} \max(f_\theta(v_j)), \qquad (5)$$

$$\pi_c = \sum_j^{|\mathcal{V}_U|} \mathbb{1}(\mathbb{I}(\max(f_\theta(v_j)) \ge \tau')), \qquad (6)$$

Since setting a threshold $\tau$ requires extensive experience and suitable thresholds may vary across datasets, we use a dynamic threshold here. With the help of the estimated class distribution of pseudo labels, we are able to balance the unsupervised loss. We utilize balanced softmax [25] here for three reasons: (1) balanced softmax can ensure Fisher consistency and achieves excellent performance in CIGL; (2) when the number of pseudo labels for some classes is zero, it remains unaffected; (3) although the current state-of-the-art performance is based on re-sampling [14, 23], the large number of pseudo labels means that adding too many virtual ego-networks, which could potentially destroy the original graph structure. Finally, the double balancing of unsupervised loss is formulated as follows:

$$\mathcal{L}_{unsup} = -\lambda \cdot \mathbb{E}_{v_j \in \mathcal{V}_U} \mathbb{I}(\max(f_\theta(v_j)) \ge \tau') \ell(\mathbf{q}_j, \hat{y}_j), \qquad (7)$$

$$\mathbf{q}_j[c] = f_\theta(v_j) + \mu \cdot \log \pi_c \qquad (8)$$

where $\mathbf{q}$ is the adjusted logits, and $\mu$ is a scaling parameter that affects the intensity of adjustment. The PyTorch-style pseudocode is presented in Algorithm 1. We can find that just a few lines of code can significantly improve the performance of CIGL with almost no additional training overhead.

Although the performance of `Double Balancing` is already excellent, potential noisy labels within the pseudo labels could harm model performance, leading to confirmation bias. To further improve results, we propose `Noise-Tolerant Double Balancing`. Inspired by Wang et al. [32], we introduce a symmetric term to facilitate noise robustness:

$$\mathcal{L}_{unsup} = -\lambda \cdot \mathbb{E}_{v_j} \mathbb{I}(\max(f_\theta(v_j)) \ge \tau')(\ell(\mathbf{q}_j, \hat{y}_j) + \beta \cdot \ell(\hat{\mathbf{y}}_j, \max(\mathbf{p}_j))), \qquad (9)$$

where $\hat{\mathbf{y}}$ is the one-hot vector of pseudo label, $\beta$ is a trade-off hyper-parameter. We simplify the subscript of expectation here.

### 3.2 Propagation then Tramsformation

While DB can leverage pseudo labels to alleviate overfitting in minority classes and effectively address label distribution shift, its effectiveness may still be suboptimal in cases where the number of labels is extremely limited. Recall Equation 4, we analyze CIGL from the perspective of domain adaptation and assume the class conditional distributions of labeled, unlabeled, and test are consistent. However, when the imbalance ratio is large, and the number of labeled nodes in minority classes is extremely limited, due to

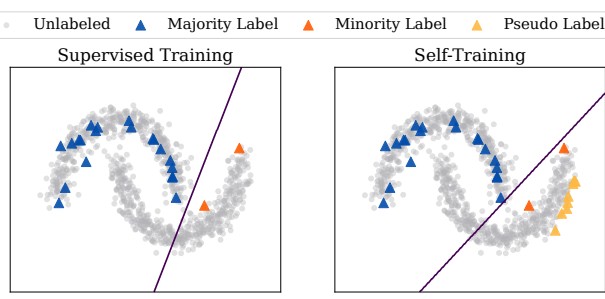

**Figure 3: Toy example on the two-moon dataset. We utilize a simple two-layer fully connected MLP for classification.**

selection bias [35], the distribution differences between the labeled set and the other two sets will increase, causing the distribution of unlabeled set to shift as well [17, 31]. Therefore, the assumption of consistent class conditional distribution does not hold, *i.e.* $p_l(\mathbf{x}|y) \ne p_u(\mathbf{x}|y) \ne p_t(\mathbf{x}|y)$.

We conduct a toy experiment on the two-moon dataset to explain the selection bias problem in an extremely limited labeled set (as shown in Figure 3). Due to the large number of labeled samples in the majority class, they uniformly distribute on the half-moon distribution and can effectively capture the ground-truth distribution. While the minority class only possesses two labeled samples, which makes it hard to cover the ground-truth distribution. Despite we use self-training to generate several pseudo labels for the minority class, they are all concentrated around the labeled samples and cannot expand their distribution. Even if the pseudo labels are all correct and the class distribution is balanced, the minority class still exhibits poorer performance compared to the majority class.

In order to figure out the selection bias in heavily imbalanced scenarios, we revisit GNNs' architecture from message-passing. Most current CIGL work is conducted on models within PTPT framework, like GCN, GAT, and GraphSAGE. However, because of the over-smoothing issue, this kind of model is hard to deepen the layers, which means labeled nodes cannot propagate supervision signals to nodes at larger propagation hops. In this work, we refer to this issue as the *Propagation Dilemma*. According to recent literature, we know that the main reason for performance degradation with increased depth may lie in the T operation rather than P operation [41]. Consequently, we attempt to use PPTT architecture in the CIGL task. In light of the strong few-shot learning ability of D$^2$PT [19], we leverage graph diffusion-based propagation here:

$$\mathbf{X}^{(t+1)} = (1 - \alpha)\tilde{\mathbf{A}}\mathbf{X}^{(t)} + \alpha\mathbf{X}, \qquad (10)$$

$$\hat{\mathbf{Y}} = \texttt{Softmax}(\texttt{MLP}(\mathbf{X}^{(T)})) \qquad (11)$$

where $\mathbf{X}^{(0)} = \mathbf{X}$, $\alpha \in (0, 1]$ is the restart probability, $T$ is the number of propagation steps, and MLP is a simple fully-connected multilayer perceptron. By increasing the number of propagation steps, we found that even without using any CIGL techniques, performance still improves with the increase in propagation hops. (as shown in Table 1). This interesting experimental result supports that few-shot is a significant challenge in heavily class-imbalanced scenarios.

**Algorithm 1:** PyTorch-style pseudocode for `D-Balancing`

```
# model: graph neural networks; get_confidence: the
  function to get confidence and predictions; M:
  existing state-of-the-art baselines; BS: balanced
  softmax function; lambda: trade-off parameter.
# Generate pseudo labels
with torch.no_grad():
    model.eval()
    logits = model(x, edge_index)
confidence, pred_label = get_confidence(logits)
# Dynamic Threshold
t = confidence[unlabel_mask].mean().item()
pseudo_mask = confidence.ge(t) & unlabel_mask
# Pseudo Label Distribution Estimation
num_list_p = [(pred_label[pseudo_mask] ==
 i).sum().item() for i in range(num_cls)]
# Existing CIGL Methods
model.train()
optimizer.zero_grad()
logits, loss = M(x, edge_index, model, train_mask)
# Double Balancing (Ours)
loss += BS(logits[pseudo_mask],
 pred_label[pseudo_mask], num_list_u) * lambda
# Backward Supervised and Unsupervised Loss
loss.backward()
optimizer.step()
```

**Table 1: In imbalanced training sets, the model's performance (Balanced Accuracy) *w.r.t.* the number of propagation hops.**

| Hop | 2 | 4 | 6 | 8 | 10 | 12 | 14 |
|---|---|---|---|---|---|---|---|
| Cora (IR=10) | 61.03 | 65.83 | 67.00 | 67.83 | 67.29 | 68.00 | 67.94 |
| Cora (IR=20) | 53.67 | 56.28 | 58.35 | 59.06 | 59.86 | 59.36 | 59.54 |
| CiteSeer (IR=10) | 44.02 | 46.91 | 49.16 | 49.41 | 49.52 | 49.81 | 49.90 |
| CiteSeer (IR=20) | 39.21 | 39.93 | 41.24 | 42.90 | 42.85 | 44.36 | 42.30 |

By combining all designs mentioned above, we propose our simple-yet-effective method, `IceBerg`, which can achieve state-of-the-art performance in class-imbalanced and few-shot scenarios. To illustrate that these modules we proposed are not simply additions, we list several advantages of `IceBerg` below:

- To propagate supervision signals to every node as much as possible, we not only use pseudo labels to increase the sources of propagation but also expand the range by increasing the number of propagation hops.
- Since parameter-free propagation can be viewed as unsupervised node representation learning, it is more robust to noisy labels and better suited for the self-training framework.
- While using a large number of pseudo labels may increase some gradient backpropagation overhead, by decoupling `P` and `T`, we can precompute node features and only need to optimize the `MLP`, which significantly reducing training costs.

## 4 EXPERIMENTS

In this section, we conduct systematic and extensive experiments to answer the following research questions:

**Table 2: Statistics of datasets**

| Dataset | #nodes | #edges | #features | #classes |
|---|---|---|---|---|
| Cora | 2,708 | 10,556 | 1,433 | 7 |
| CiteSeer | 3,327 | 9,104 | 3,703 | 6 |
| PubMed | 19,717 | 88,648 | 500 | 3 |
| CS | 18,333 | 163,788 | 6,805 | 15 |

- **RQ1**: How does `IceBerg` perform with respect to diverse datasets and CIGL baselines?
- **RQ2**: Compared to other Few-Shot Graph Learning (FSGL) baselines, can `IceBerg` achieve better performance?
- **RQ3**: How does each component and hyper-parameter influence the model performance?
- **RQ4**: How efficient is our model in terms of training time?

### 4.1 Experimental settings

*4.1.1 Benchmark datasets.* We conduct experiments on three widely used citation network datasets, Cora, CiteSeer, and PubMed, and one co-author network dataset, Coauthor-CS. The statistics of four datasets can be found in Table 2. We utilize the public splits in Yang et al. [38] for Cora, CiteSeer, and PubMed. For Coauthor-CS, since there is no public split, we randomly select 20 nodes per class for training, 30 nodes per class for validation, and the rest for testing. Following Park et al. [23] and Song et al. [27], we construct imbalanced training sets by removing labeled nodes from the balanced training sets. Specifically, we select minority classes as half the number of classes ($C/2$) and alter labeled nodes of minority classes to unlabeled nodes randomly until the number of nodes in each minority class equals the ratio of the number of majority nodes in the most frequent class to imbalance ratio ($N_L^1/\gamma_L$). In this paper, we adopt imbalance ratios of 10 and 20. A more intuitive class distribution is shown in Figure 2. And we also conduct few-shot experiments on three citation network graphs. We randomly select 1, 2, 3, and 5 labels for each class from the whole graph, and select 30 labels per class for validation, and the rest for testing.

*4.1.2 Baselines.* Since we evaluate the effectiveness of our proposed `IceBerg` in class-imbalanced and few-shot node classification scenarios, we select 17 baselines from the two branches: **(I) CIGL baselines:** This branch can be further divided into three categories: (i) Loss function-oriented, Re-Weight (RW) [11] simply up-weights the minority classes and down-weight the majority classes in loss function according to quantity; BalancedSoftmax (BS) [25] accommodate the label distribution shift between training and testing; ReNode (RN) [2] alleviate the topology imbalance by re-weighting the influence of labeled nodes based on their position. (ii) Re-sampling-oriented, Mixup (MIX) utilizes mixup to generate minority classes and duplicate neighborhoods; GraphENS (ENS) [23] synthesizes the whole ego networks for minority classes; GraphSHA (SHA) [14] aims to synthesize harder minority samples. (iii) Topology-aware adjustment, TAM [27] adaptively adjusts the margin according to connectivity pattern, BAT [21] augments topology to address ambivalent and distant message-passing. **(II) FSGL baselines:** we select two categories for this branch: (i) decouple GNNs, such as APPNP [8], SGC [33], DAGNN [18], and $D^2PT$ [19].

(ii) Semi-supervised approaches, such as Self-Training [13], M3S [28], CGPN [29], Meta-PN [5], and DR-GST [17].

*4.1.3 Evaluation metrics.* To ensure a comprehensive and fair evaluation, we utilize Balanced Accuracy and Macro-F1 as metrics for class-imbalanced node classification. We use Accuracy to evaluate the performance of few-shot node classification.

*4.1.4 Implementation details.* We use hyper-parameters that the authors provided for baselines. For experiments where the authors did not provide hyper-parameter settings, we conducted simple tuning. Besides, we integrate all CIGL baselines in our experimental framework for a fairer comparison. All the models are implemented by PyTorch version 2.0.1 with PyTorch Geometric version 2.3.1. All experiments are conducted on Nvidia GeForce RTX 3090.

## 4.2 RQ1: Performance comparison in CIGL

To answer **RQ1** and verify the effectiveness of our proposed `IceBerg` in CIGL, we use seven baselines as base balancing methods, adding two topology-aware adjustment plugins to all base balancing methods, along with our proposed `Double Balancing` and `IceBerg`. Table 3 presents the results on four benchmark datasets with an imbalanced ratio of 10. From the Table 3, our observations can be threefold: (1) We observe that on all datasets, TAM and BAT as plugins can enhance the performance of base balancing methods. However, DB, with its straightforward idea and implementation, easily and significantly outperforms both TAM and BAT. (2) `IceBerg` enhances the model's few-shot capability by decoupling propagation and transformation operations, improving training efficiency while further boosting the performance of DB. (3) Since BAT uses virtual super nodes as shortcuts to propagate supervision signals to more distant nodes, it can also be viewed as a method of augmenting supervision signals. Given that both BAT and DB achieve significant improvement on CiteSeer, we infer that the few-shot issue may be more severe on CiteSeer. Additionally, we evaluated under a more imbalanced experimental setting (as shown in Table 5) and found that `IceBerg`'s excellent few-shot capability results in even more noticeable improvements.

## 4.3 RQ2: Performance comparison in FSGL

We also conduct experiments on Cora, CiteSeer, and PubMed datasets in the few-shot node classification task. We randomly select 1, 2, 3, and 5 labels for each class. The experimental results can be found in Table 4. We can observe that: (1) Even in a balanced training dataset, the *Matthew Effect* may still occur, yet DB continues to perform exceptionally well in balanced few-shot datasets. (2) Since decoupling GNNs are able to propagate supervision signals to more distant nodes, `IceBerg` can further enhance performance on top of DB. (3) `IceBerg` shows even more significant performance improvements in scenarios with extremely scarce labels (*e.g.* 1 label per class), which further validates its excellent few-shot learning capability.

## 4.4 RQ3: Ablation and parameter study

To explore **RQ3**, we conducted extensive ablation studies, as shown in Table 3, Table 4, and Table 5. We evaluated across different imbalance ratios on four datasets and found that `IceBerg` consistently

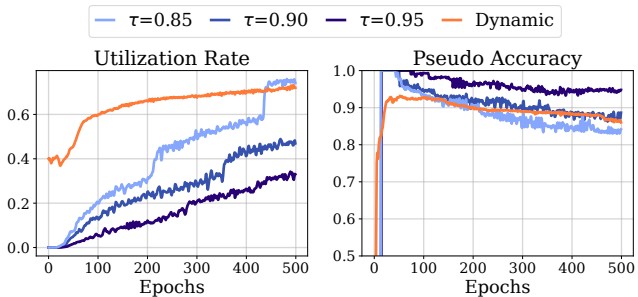

Figure 4: On the Cora dataset, when the imbalance ratio (IR) is 20, the utilization rate of unlabeled nodes and the accuracy of pseudo labels vary across epochs under different thresholds.

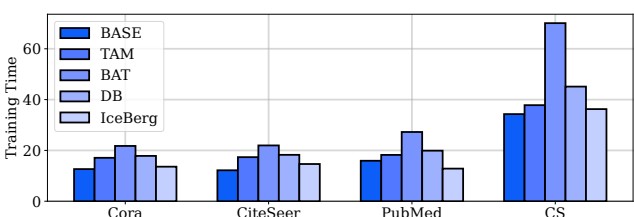

Figure 5: The time it takes to run 1000 epochs across datasets.

outperforms DB. Additionally, to study the trade-off between quantity and quality of pseudo labels, we compare our proposed dynamic threshold with predefined fixed thresholds. The experimental results are shown in Figure 4. It is obvious that dynamic threshold can leverage a large number of pseudo labels in the early training stage (when the training epoch reaches 100, it can utilize nearly 60% unlabeled nodes) while still maintaining a high level of accuracy (about 90% prediction accuracy of pseudo labels).

## 4.5 RQ4: Efficiency study

To answer **RQ4**, we run 1000 epochs for base balancing methods, TAM, BAT, DB, and `IceBerg` on different datasets. As shown in Figure 5, TAM and DB will just provide a little training overhead compared to BASE, while BAT may take twice as long on some datasets as BASE. Considering the performance enhancement of DB, it can be viewed as a free lunch in CIGL and FSGL. Furthermore, since `IceBerg` precomputes the propagation process before training, it can largely reduce the training costs. It can achieve even better efficiency on some datasets than BASE.

## 4.6 Visualization

To better understand the balancing ability of our propose DB and `IceBerg`, we visualize the learned node representations on the Cora dataset with imbalance ratio equal to 10 in Figure 6. We select BalancedSoftmax as the base balancing technique. And we can observe that DB and `IceBerg` show better class boundaries and less distribution overlapping compared to other methods, which further proves the effectiveness of our methods.

**Table 3: Model performance on benchmark graph datasets. We report balanced accuracy and macro-f1 of each independent CIGL baselines *w.r.t.* various plug-and-play modules. The best average results are highlighted in bold. Each experimental result is obtained from the mean of 10 repeated experiments. OOT stands for out-of-time, *i.e.* the time exceeds one day.**

| Metric | | Balance Acc. (↑) | | | | | | | | Macro-F1 (↑) | | | | | | | |
|---|---|---|---|---|---|---|---|---|---|---|---|---|---|---|---|---|---|
| Baselines | | ERM | RW | BS | RN | MIX | ENS | SHA | Avg. (Δ) | ERM | RW | BS | RN | MIX | ENS | SHA | Avg. (Δ) |
| Cora | BASE | 60.95 | 65.52 | 68.46 | 67.61 | 65.49 | 70.31 | 73.13 | 67.35 | 59.30 | 65.54 | 68.41 | 67.27 | 65.70 | 70.31 | 72.78 | 67.04 |
| | +TAM | 61.63 | 67.25 | 69.90 | 67.18 | 69.96 | 71.52 | 73.22 | 68.66 (+1.31) | 59.69 | 66.76 | 68.41 | 67.39 | 68.18 | 71.71 | 72.89 | 67.86 (+0.82) |
| | +BAT | 69.80 | 72.14 | 70.74 | 71.84 | 71.57 | 72.58 | 74.46 | 71.87 (+4.52) | 68.68 | 70.31 | 69.53 | 70.59 | 70.93 | 72.28 | 73.30 | 70.80 (+3.76) |
| | +DB | 70.14 | 73.60 | 75.20 | 74.22 | 75.11 | 76.07 | 75.08 | 74.20 (+6.85) | 68.97 | 72.96 | 74.07 | 73.14 | 74.05 | 75.32 | 74.85 | 73.33 (+6.29) |
| | +IceBerg | 75.04 | 75.91 | 76.69 | 76.86 | 75.78 | 74.78 | 74.90 | **75.70** (+8.35) | 74.55 | 75.93 | 75.80 | 76.45 | 75.27 | 74.78 | 73.93 | **75.24** (+8.20) |
| CiteSeer | BASE | 38.21 | 44.52 | 53.70 | 47.78 | 47.10 | 55.42 | 57.34 | 49.15 | 29.40 | 38.85 | 50.73 | 42.51 | 42.00 | 53.85 | 54.99 | 44.61 |
| | +TAM | 43.23 | 43.23 | 55.54 | 48.36 | 50.55 | 57.47 | 59.50 | 51.12 (+1.97) | 35.19 | 39.31 | 54.18 | 42.48 | 45.61 | 56.23 | 58.22 | 47.31 (+2.70) |
| | +BAT | 55.37 | 58.36 | 60.86 | 59.28 | 59.76 | 62.67 | 63.40 | 59.95 (+10.80) | 54.94 | 57.54 | 60.01 | 58.07 | 57.79 | 62.46 | 62.61 | 59.06 (+14.45) |
| | +DB | 59.55 | 61.92 | 65.87 | 65.12 | 61.47 | 65.23 | 62.66 | 63.11 (+13.96) | 56.96 | 59.53 | 64.65 | 64.18 | 58.63 | 64.77 | 60.97 | 61.38 (+16.77) |
| | +IceBerg | 63.95 | 65.65 | 64.73 | 65.28 | 61.51 | 64.25 | 63.95 | **64.18** (+15.03) | 62.71 | 65.12 | 63.98 | 64.53 | 58.28 | 63.37 | 61.84 | **62.83** ±18.22 |
| PubMed | BASE | 65.21 | 70.17 | 72.97 | 71.52 | 72.92 | 71.89 | 74.92 | 71.37 | 55.43 | 66.37 | 70.80 | 67.86 | 71.40 | 71.07 | 73.92 | 68.12 |
| | +TAM | 68.54 | 70.01 | 74.13 | 71.00 | 73.95 | 74.01 | 76.13 | 72.53 (+1.16) | 62.96 | 66.75 | 73.27 | 67.18 | 73.08 | 72.41 | 75.31 | 70.13 (+2.01) |
| | +BAT | 67.57 | 73.37 | 74.86 | OOT | 73.23 | 76.91 | 75.34 | 73.54 (+2.17) | 64.40 | 73.24 | 73.34 | OOT | 71.24 | 76.86 | 74.73 | 72.30 (+4.18) |
| | +DB | 75.39 | 78.02 | 77.59 | 77.90 | 78.08 | 75.99 | 77.27 | 77.17 (+5.80) | 71.09 | 75.98 | 76.16 | 76.03 | 76.21 | 74.64 | 75.76 | 75.12 (+7.00) |
| | +IceBerg | 78.49 | 77.96 | 76.85 | 78.16 | 78.36 | 76.12 | 76.43 | **77.48** (+6.11) | 75.98 | 75.59 | 74.92 | 76.67 | 76.44 | 74.38 | 74.72 | **75.52** (+7.40) |
| CS | BASE | 74.95 | 80.07 | 84.16 | 80.30 | 81.47 | 84.60 | 86.49 | 81.72 | 70.20 | 77.78 | 83.08 | 78.10 | 79.74 | 83.45 | 84.92 | 79.61 |
| | +TAM | 74.61 | 80.68 | 84.28 | 80.87 | 81.87 | 85.44 | 87.26 | 82.11 (+0.39) | 69.89 | 78.72 | 82.72 | 78.61 | 80.05 | 84.81 | 86.13 | 80.13 (+0.52) |
| | +BAT | 83.42 | 87.32 | 87.29 | OOT | 85.82 | 88.73 | 88.40 | 86.83 (+5.11) | 78.03 | 85.02 | 83.60 | OOT | 83.41 | 86.95 | 86.38 | 83.89 (+4.28) |
| | +DB | 87.21 | 87.59 | 89.34 | 89.00 | 89.47 | 89.02 | 89.39 | **88.71** (+6.99) | 83.85 | 84.84 | 87.58 | 86.68 | 88.57 | 87.07 | 87.58 | **86.59** (+6.98) |
| | +IceBerg | 86.97 | 88.24 | 88.15 | 88.58 | 89.04 | 89.26 | 89.11 | 88.47 (+6.75) | 82.75 | 86.02 | 87.00 | 86.56 | 88.51 | 87.29 | 87.43 | 86.50 (+6.89) |

**Table 4: Model performance on Cora, CiteSeer, and Pubmed with 1,2,3, and 5 labeled nodes per class. We report the classification accuracy of each method. The best results are highlighted in bold. Each experimental result is obtained from the mean of 10 repeated experiments.**

| Dataset | Cora | | | | CiteSeer | | | | PubMed | | | |
|---|---|---|---|---|---|---|---|---|---|---|---|---|
| L/C | 1 | 2 | 3 | 5 | 1 | 2 | 3 | 5 | 1 | 2 | 3 | 5 |
| GCN | 46.6±2.3 | 58.8±1.3 | 63.5±1.8 | 68.5±1.2 | 37.4±2.3 | 49.8±1.2 | 53.8±1.8 | 59.5±1.1 | 55.9±2.7 | 57.6±3.0 | 61.81±2.1 | 68.7±1.5 |
| APPNP | 51.9±2.8 | 63.0±2.5 | 66.8±2.0 | 73.3±0.8 | 35.8±2.1 | 48.9±1.6 | 52.6±1.7 | 58.1±0.7 | 57.4±3.5 | 59.1±2.8 | 63.6±2.2 | 70.4±1.6 |
| SGC | 51.0±3.0 | 59.2±3.5 | 64.0±3.0 | 70.3±2.1 | 34.4±2.1 | 47.1±3.1 | 50.4±2.5 | 60.4±1.3 | 55.3±3.7 | 57.0±2.7 | 58.5±2.0 | 63.5±2.4 |
| DAGNN | 54.3±2.0 | 62.8±1.8 | 67.3±1.5 | 73.4±0.6 | 44.6±1.8 | 53.2±1.7 | 56.9±1.4 | 59.9±1.2 | 57.2±3.9 | 60.1±3.2 | 61.6±3.3 | 68.4±3.2 |
| $D^2PT$ | 62.6±6.9 | 69.6±8.8 | 74.1±6.8 | 76.8±4.4 | 54.5±12.8 | 64.3±4.0 | 66.5±1.2 | 67.4±0.9 | 59.9±10.4 | **65.7±5.3** | 69.3±4.7 | 72.5±2.6 |
| S-Training | 52.6±7.9 | 61.8±7.2 | 66.5±6.7 | 76.2±2.1 | 31.6±7.6 | 47.8±4.4 | 50.3±5.8 | 57.8±4.9 | 55.6±6.8 | 61.5±6.5 | 65.1±7.3 | 69.5±4.7 |
| M3S | 50.7±7.4 | 61.1±5.0 | 70.1±3.5 | 76.6±1.8 | 38.7±9.6 | 44.6±8.0 | 57.4±6.8 | 63.7±6.5 | 55.4±10.1 | 67.2±4.1 | 70.2±4.7 | 69.7±3.3 |
| CGPN | 64.3±9.1 | 63.8±9.0 | 68.3±3.6 | 71.1±1.8 | 49.4±9.4 | 53.3±5.2 | 54.1±4.3 | 57.0±5.6 | 56.7±6.3 | 60.1±8.0 | 66.9±3.4 | 65.9±4.2 |
| Meta-PN | 55.8±3.3 | 72.7±2.1 | 74.6±2.0 | 76.4±1.3 | 34.8±4.8 | 42.6±3.6 | 56.2±1.9 | 59.8±4.0 | 54.4±0.0 | 63.4±1.6 | **69.6±0.6** | **73.6±1.6** |
| DR-GST | 50.1±11.3 | 62.3±7.7 | 68.9±7.1 | 76.1±5.1 | 42.9±9.4 | 53.1±4.5 | 57.8±5.9 | 63.7±2.9 | 56.3±9.9 | 61.5±9.7 | 63.7±4.3 | 69.7±5.4 |
| DB | 57.7±2.7 | 68.0±1.3 | 70.9±1.5 | 74.9±1.2 | 53.8±4.2 | 64.6±1.7 | 66.1±1.5 | **68.2±0.4** | 55.4±4.2 | 58.8±3.4 | 62.0±3.1 | 68.5±3.2 |
| IceBerg | **68.2±2.4** | **75.1±1.1** | **75.8±1.1** | **78.3±0.7** | **57.8±2.5** | **64.8±2.0** | **66.6±1.0** | 67.8±0.6 | **60.1±3.2** | 63.7±3.2 | 64.7±2.6 | 69.4±2.6 |

## 5   RELATED WORK

This work is related to two research fields: Class-Imbalanced Graph Learning (CIGL) and Few-Shot Graph Learning (FSGL).

### 5.1   Class-Imbalanced Graph Learning

We will first introduce the related work in CIGL. Due to the GNNs inheriting the character of deep neural networks, GNNs perform with biases toward majority classes when training on imbalanced datasets. To overcome this challenge, CIGL has emerged as a promising solution that combines the strengths of graph representation learning and class-imbalanced learning. A great branch of this field is oversampling minority nodes by data augmentation to balance the skew training label distribution. GraphSMOTE [42] leverages representative data augmentation method (*i.e.*, SMOTE) and proposes edge predictor to fuse augmented nodes into the original graph. GraphENS [23] discovers neighbor memorization phenomenon in imbalanced node classification, and generates minority nodes by synthesizing ego-networks according to similarity. GraphSHA [14] only synthesizes harder training samples and blocks message-passing from minority nodes to neighbor classes by generating connected edges from 1-hop subgraphs. Apart from that, some methods aim to facilitate CIGL through the graph structure. TAM [27] adjusts margins node-wisely according to the extent of deviation from connectivity patterns. BAT [21] is a data augmentation approach, which alleviates ambivalent and distant message-passing in imbalanced node classification. Since the numerous real-world

**Table 5: Model performance on benchmark graph datasets with heavy class-imbalanced. We report balanced accuracy and macro-f1 of each independent CIGL baselines *w.r.t.* various plug-and-play modules. The best average results are highlighted in bold. Each experimental result is obtained from the mean of 10 repeated experiments.**

| Metric | | Balance Acc. (↑) | | | | | | | | Macro-F1 (↑) | | | | | | | |
|---|---|---|---|---|---|---|---|---|---|---|---|---|---|---|---|---|---|
| | Baselines | ERM | RW | BS | RN | MIX | ENS | SHA | Avg. (Δ) | ERM | RW | BS | RN | MIX | ENS | SHA | Avg. (Δ) |
| **Cora** | BASE | 53.18 | 59.08 | 64.16 | 59.57 | 59.92 | 64.13 | 69.99 | 61.43 | 48.15 | 56.60 | 63.38 | 58.24 | 57.57 | 64.13 | 69.98 | 59.72 |
| | +TAM | 55.50 | 62.15 | 65.34 | 59.15 | 62.87 | 64.13 | 70.66 | 62.82 (+1.39) | 50.95 | 59.54 | 65.19 | 57.56 | 61.29 | 64.14 | 70.91 | 61.36 (+1.64) |
| | +BAT | 72.19 | 70.55 | 68.41 | 73.36 | 69.95 | 73.14 | 74.24 | 71.69 (+10.26) | 71.07 | 69.60 | 67.29 | 72.22 | 69.04 | 72.33 | 73.16 | 70.67 (+10.95) |
| | +DB | 70.63 | 71.96 | 71.43 | 71.20 | 72.80 | 72.18 | 75.50 | 72.24 (+10.81) | 68.17 | 70.19 | 70.51 | 69.35 | 71.40 | 71.29 | 74.32 | 70.74 (+11.02) |
| | +IceBerg | 68.41 | 75.10 | 75.41 | 76.24 | 73.57 | 73.94 | 75.45 | **74.01** (+12.58) | 67.21 | 74.59 | 74.71 | 75.50 | 72.50 | 72.38 | 74.11 | **73.00** (+13.28) |
| **CiteSeer** | BASE | 34.67 | 40.45 | 46.43 | 41.26 | 39.30 | 45.46 | 50.77 | 42.62 | 23.04 | 33.68 | 41.98 | 33.19 | 30.71 | 40.21 | 47.17 | 35.71 |
| | +TAM | 37.63 | 42.79 | 48.22 | 42.09 | 40.18 | 47.01 | 52.08 | 44.28 (+1.66) | 27.09 | 35.51 | 44.45 | 36.10 | 32.51 | 41.65 | 48.98 | 38.04 (+2.33) |
| | +BAT | 54.68 | 59.29 | 47.77 | 62.80 | 53.27 | 62.41 | 59.75 | 57.13 (+14.5) | 50.03 | 57.16 | 46.84 | 61.49 | 48.79 | 61.59 | 58.47 | 54.91 (+19.20) |
| | +DB | 45.98 | 59.20 | 54.09 | 56.29 | 55.72 | 61.18 | 54.37 | 55.26 (+12.64) | 39.35 | 56.93 | 50.74 | 54.29 | 51.63 | 60.64 | 50.82 | 52.05 (+16.34) |
| | +IceBerg | 51.01 | 59.06 | 65.88 | 60.57 | 57.14 | 62.45 | 58.26 | **59.19** (+16.57) | 44.75 | 55.91 | 64.40 | 58.01 | 53.26 | 61.97 | 55.25 | **56.22** (+20.51) |
| **PubMed** | BASE | 61.72 | 66.28 | 68.46 | 67.63 | 65.96 | 68.67 | 72.72 | 67.34 | 47.81 | 58.53 | 66.89 | 59.29 | 62.71 | 65.22 | 71.23 | 61.66 |
| | +TAM | 63.42 | 68.25 | 69.86 | 67.10 | 68.52 | 69.83 | 73.18 | 68.59 (+1.25) | 54.12 | 64.82 | 69.82 | 62.30 | 64.34 | 66.19 | 72.15 | 64.82 (+3.16) |
| | +BAT | 71.42 | 71.74 | 72.39 | OOT | 73.12 | 73.77 | 71.56 | 72.33 (+4.99) | 67.96 | 69.39 | 71.59 | OOT | 70.11 | 72.14 | 69.41 | 70.10 (+8.44) |
| | +DB | 78.10 | 77.42 | 75.51 | 76.53 | 76.55 | 76.22 | 74.33 | **76.38** (+9.04) | 75.39 | 74.86 | 72.99 | 74.12 | 74.59 | 73.42 | 72.15 | 73.93 (+12.27) |
| | +IceBerg | 76.94 | 77.02 | 76.48 | 76.31 | 76.45 | 76.83 | 74.13 | 76.30 (+8.96) | 74.00 | 74.49 | 74.06 | 73.64 | 74.75 | 74.02 | 72.79 | **73.96** (+12.30) |
| **CS** | BASE | 63.86 | 72.46 | 77.24 | 71.87 | 73.45 | 78.90 | 83.79 | 74.51 | 55.45 | 67.39 | 75.29 | 67.79 | 68.94 | 76.54 | 80.91 | 70.33 |
| | +TAM | 67.82 | 74.99 | 78.22 | 74.18 | 75.98 | 80.21 | 84.41 | 76.54 (+2.03) | 61.67 | 71.35 | 76.92 | 71.51 | 73.02 | 78.97 | 82.32 | 73.68 (+3.35) |
| | +BAT | 77.32 | 85.24 | 84.17 | OOT | 83.08 | 88.07 | 87.65 | 84.25 (+9.74) | 70.66 | 82.52 | 79.30 | OOT | 79.53 | 85.75 | 85.43 | 80.53 (+10.20) |
| | +DB | 79.45 | 83.27 | 85.16 | 85.47 | 85.27 | 86.35 | 86.67 | **84.52** (+10.01) | 72.32 | 78.40 | 82.60 | 81.75 | 81.15 | 83.79 | 84.11 | **80.58** (+10.25) |
| | +IceBerg | 81.10 | 82.98 | 84.46 | 84.97 | 85.36 | 85.13 | 86.88 | 84.41 (+9.90) | 74.37 | 76.49 | 82.49 | 81.54 | 81.67 | 82.57 | 84.42 | 80.50 (+10.17) |

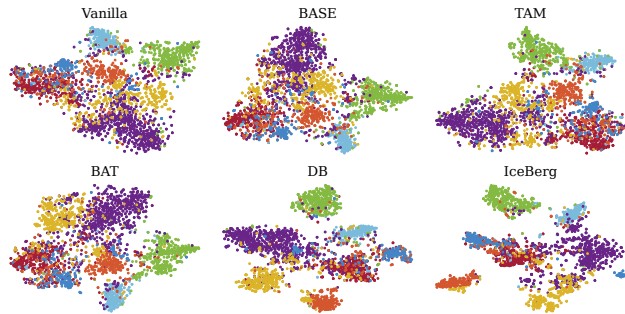

**Figure 6: Visualization of node representations.**

applications of CIGL, its techniques have been applied to many other tasks as well, *e.g.*, graph anomaly detection [22, 44], graph fairness learning [15], and graph contrastive learning [39].

## 5.2 Few-Shot Graph Learning

Besides, modern artificial intelligence is heavily dependent on a large number of high quality labels. Considering complexity and heterogeneity of graph-structured data, human labeling is unbearably laborious. Therefore, there are some works that aim to figure out the few-shot issues in graph machine learning. DAGNN [18] and D2PT [19] disentangle propagation and transformation to transmit the supervision signals to more distant nodes. Additionally, D2PT utilizes dual-channel contrastive learning to enhance its capability of capturing unsupervised knowledge. And self-training is also a promising technology to alleviate label scarcity with the help of pseudo labels. Liu et al. [17] find that high-confidence pseudo labels

may introduce distribution shift, so they reweigh the loss function by information gain. M3S [28] leverages the DeepClustering technique to refine the self-training process. Meta-PN [5] utilizes meta-learning label propagation to construct a pseudo label set and decouple the model architecture to allow larger receptive fields.

Although existing literature have achieved considerable success in CIGL and FSGL, there is no work analyzes the connection between the two research fields. In this work, we first time study the Matthew effect challenge in CIGL and FSGL, and achieve the state-of-the-art performance in two fields with one unified framework.

## 6 CONCLUSION

In this paper, we provide a key statement that the labeled nodes in the graph are just the tip of the iceberg, and if we could effectively utilize a large number of unlabeled nodes, we can significantly and easily achieve state-of-the-art performance. We first study the Matthew Effect in the self-training framework, and based on theoretical analysis, we propose `Double Balancing`. We find that balancing the pseudo label set can nicely alleviate the Matthew Effect and enhance performance. To avoid Confirmation Bias, we propose Noise-Tolerant Double Balancing. Additionally, in the heavy class-imbalanced scenarios, minority classes may also face few-shot issues. Therefore, we disentangle the propagation and transformation operations for augmenting supervision signals to distant nodes. Combining all the above designs, we propose `IceBerg`, a simple-yet-effective approach to class-imbalanced and few-shot node classification. It can achieve excellent performance with good efficiency on various benchmark datasets. At last, we suggest that future research works pay more attention to the large number of unlabeled nodes present in the graph, rather than just treat CIGL or FSGL tasks as supervised learning tasks.

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
