# OpenReview forum: "Beyond tip of the Iceberg: Debiased Self-training for Long-tailed Semi-supervised Node Classification"
_ACM.org/TheWebConf/2025/Conference — WWW 2025 Poster_

### Official Review · Reviewer_1Miq · 2024-11-01

**Novelty:** 3
**Technical Quality:** 4

**Review:**

# Summary

* The authors introduce IceBerg, a framework that addresses the challenges of class-imbalanced and few-shot learning in Graph Neural Networks (GNNs). By leveraging unlabeled nodes through a debiased self-training approach, IceBerg improves GNN performance in semi-supervised scenarios.

# Strenght

* The writing and motivation is very clear (e.g., Matthew effect in Fig. 1 and empirical analysis in Fig. 2)

* Several theoretical guarantees render the proposed method convincing.

* The authors conduct various experiments, including the balanced / classification accuracy, ablation study, and visualization.

# Weaknesses

* In Figure 2, the author claims that the labeled set is highly imbalanced. Nevertheless, the citation graph datasets (Cora, Citeseer, and PubMed) offer k-fold cross-validation samples, which can alleviate this issue. Moreover, random sampling of the training sets can also mitigate the imbalance problem. From my perspective, the motivation appears somewhat exaggerated.

* The concept of class-imbalanced semi-supervised learning has already been proposed in [1]. The authors need to cite this paper and clearly introduce the main differences between their work and the existing research.
  * Rethinking semi-supervised imbalanced node classification from bias-variance decomposition, NeurIPS '24

* The experiments are mostly conducted on homophilic datasets. However, the author needs to verify the effectiveness of the proposed method in heterophilic settings, where a balanced class distribution may not necessarily lead to improved performance.

**Questions:**

Please see the weaknesses above

**Reviewer Confidence:**

3: The reviewer is confident but not certain that the evaluation is correct

**Scope:**

4: The work is relevant to the Web and to the track, and is of broad interest to the community

---

### Official Review · Reviewer_oYRm · 2024-11-26

**Novelty:** 5
**Technical Quality:** 6

**Review:**

The authors propose **IceBerg**, a debiased self-training framework to address the class-imbalanced and few-shot cases for GNNs.

## Pros
- The proposed method demonstrates the best performance in the considered problem.
- The intuitions behind the proposed ideas are clearly explained and well-supported.
- The authors conducted extensive experiments.

## Cons
- The datasets used in the experiments are too small to draw generalizable conclusions.
- The presentation of the paper needs improvement.

**Questions:**

**Q1.** Regarding the datasets, the size of the data seems too small. For commonly used datasets like Cora, Citeseer, and Pubmed, it is unclear whether the model achieves good performance due to its novel ideas or simply because overfitting is less likely to occur. Could the authors clarify this distinction?

**Q2.** The equations in Section 3.1 are initially difficult to understand. Adding subscripts to the expectations is necessary for both clarity and correctness. Additionally, the explanation for the statement on line 324, "Since the target test class distribution is balanced," is insufficient in the earlier sections, making it harder to grasp. Finally, as a suggestion, the pseudocode in Algorithm 1 would benefit from being written in a more standard pseudocode format.

**Reviewer Confidence:**

2: The reviewer is willing to defend the evaluation, but it is likely that the reviewer did not understand parts of the paper

**Scope:**

3: The work is somewhat relevant to the Web and to the track, and is of narrow interest to a sub-community

---

### Official Review · Reviewer_DnZM · 2024-11-29

**Novelty:** 4
**Technical Quality:** 4

**Review:**

### Summary

This paper focuses on addressing the class-imbalanced and few-shot challenges for GNNs and proposes a debiased self-training framework called Iceberg. Specifically, the framework include Double Balancing and disentangling the propagation and transformation operations of GNNs. Comprehensive experiments can well support the effectiveness of the framework.

### Strengths

1. The topic of class-imbalanced and few-shot challenges for GNNs is interesting and has research value.
1. The paper organization is good, and the writing is well-polished.
1. Comprehensive experiments are conducted, which can well support the effectiveness and efficiency of the framework.

### Weaknesses

1.  Some details in the methodology are somewhat confusing. For example, how is Eq.(8) computed? As $f_{\theta}(v_j)$ is a vector and $\pi_c$ is a single value, which entry does the add operation perform on? Besides, the notation $\textbf{p}_j$ in Eq.(9) is used without clarification. Furthermore, the motivation of Noise-Tolerant Double Balancing is not well explained. It would be better to provide some intuitive thinking or theoretical analysis.
1.  Hyperparameters experiments are missing.
1.  Some presentation can be further improved. For example,  the word "current" is misspelled as "corrent" in line 441. Besides, as the framework Iceberg consists of two parts, it's suggested to also provide a pseudo code for the entire framework for better understanding.

**Questions:**

Please refer to the Weaknesses above.

**Reviewer Confidence:**

3: The reviewer is confident but not certain that the evaluation is correct

**Scope:**

4: The work is relevant to the Web and to the track, and is of broad interest to the community

---

### Official Review · Reviewer_uY25 · 2024-12-02

**Novelty:** 3
**Technical Quality:** 6

**Review:**

The authors introduced IceBerg, a novel method designed to address the class imbalance problem in few-shot settings. At the core of their approach is the double balance technique, which dynamically adjusts the confidence threshold for unlabeled nodes in the unsupervised loss term, effectively mitigating the Mathew effect. Additionally, they implemented a propagate-and-transform message passing mechanism to prevent distribution shift in few-shot scenarios. The effectiveness of their method was demonstrated through improved classification performance when applied to several Class Imbalanced Graph Learning (CIGL) baselines, incorporating both the double balance technique and IceBerg. Furthermore, the authors showcased the superiority of their approach in few-shot scenarios compared to various Graph Neural Network (GNN) baselines.

Here's a corrected version of your text with improved grammar:

Strengths:
1. The presentation is easy to follow and understand.
2. They demonstrated many empirical proofs to show the problem of inconsistent distribution among labeled nodes, unlabeled nodes, and testing nodes, and illustrated how these factors impact the test error.
3. The proposed method can be easily incorporated into many existing methods[1][2].

Please check questions for weakness.

**Questions:**

1. Can IceBerg still work when the size of graph is large? I have noticed that the graphs for experiments are relatively small. It would be great if you can show how the performance changes when the size of graph is large.

2. Although the authors have shown that adding DB can improved performance, it's still unclear why adding DB can mitigate the problem of class imbalance.

3. Why IceBerg and DB have lower performance in PubMed compared to baselines in Table 4?

**Reviewer Confidence:**

3: The reviewer is confident but not certain that the evaluation is correct

**Scope:**

4: The work is relevant to the Web and to the track, and is of broad interest to the community

---

### Official Review · Reviewer_EiqH · 2024-12-04

**Novelty:** 4
**Technical Quality:** 4

**Review:**

This paper improves self-training on graphs with a simple yet effective imbalance learning approach.

Pros:
- it conducts a set of analysis and found that 1st round of pseudo labels can serve as an estimator of class distributions
- it revises self-training objective with a D-balancing approach, which can alleviate the imbalance during learning pseudo-labeled nodes. And a simplified prop-transform layer architecture is used.
- experiments verify its effectiveness.

Cons:
- The baseline implementation is a bit unclear. For example, what is the network architecture of BASE in table 3?

**Questions:**

Please refer to weakness part.

**Reviewer Confidence:**

3: The reviewer is confident but not certain that the evaluation is correct

**Scope:**

4: The work is relevant to the Web and to the track, and is of broad interest to the community